# IFN-Gamma and TNF-Alpha as a Priming Strategy to Enhance the Immunomodulatory Capacity of Secretomes from Menstrual Blood-Derived Stromal Cells

**DOI:** 10.3390/ijms222212177

**Published:** 2021-11-10

**Authors:** María Ángeles de Pedro, María Gómez-Serrano, Federica Marinaro, Esther López, María Pulido, Christian Preußer, Elke Pogge von Strandmann, Francisco Miguel Sánchez-Margallo, Verónica Álvarez, Javier G. Casado

**Affiliations:** 1Stem Cell Therapy Unit, Jesús Usón Minimally Invasive Surgery Centre, 10071 Cáceres, Spain; madepedro@ccmijesususon.com (M.Á.d.P.); fmarinaro@ccmijesususon.com (F.M.); mpulido@ccmijesususon.com (M.P.); valvarez@ccmijesususon.com (V.Á.); 2Institute for Tumor Immunology, Center for Tumor Biology and Immunology (ZTI), Philipps University, 35043 Marburg, Germany; maria.gomezserrano@imt.uni-marburg.de (M.G.-S.); preusserc@staff.uni-marburg.de (C.P.); elke.poggevonstrandmann@imt.uni-marburg.de (E.P.v.S.); 3CIBER de Enfermedades Cardiovasculares (CIBERCV), 28029 Madrid, Spain; jgarcas@unex.es; 4Immunology Unit, University of Extremadura, 10003 Cáceres, Spain; 5Institute of Molecular Pathology Biomarkers, University of Extremadura, 10003 Caceres, Spain

**Keywords:** menstrual blood, mesenchymal stromal cells, secretome, priming, extracellular vesicles

## Abstract

Mesenchymal stromal cells isolated from menstrual blood (MenSCs) exhibit a potent pro-angiogenic and immunomodulatory capacity. Their therapeutic effect is mediated by paracrine mediators released by their secretomes. In this work, we aimed to evaluate the effect of a specific priming condition on the phenotype and secretome content of MenSCs. Our results revealed that the optimal condition for priming MenSCs was the combination of interferon gamma (IFNγ) and tumor necrosis factor alpha (TNFα) that produced a synergistic and additive effect on IDO1 release and immune-related molecule expression. The analyses of MenSC-derived secretomes after IFNγ and TNFα priming also revealed an increase in EV release and in the differentially expressed miRNAs involved in the immune response and inflammation. Proliferation assays on lymphocyte subsets demonstrated a decrease in CD4+ T cells and CD8+ T cells co-cultured with secretomes, especially in the lymphocytes co-cultured with secretomes from primed cells. Additionally, the expression of immune checkpoints (PD-1 and CTLA-4) was increased in the CD4+ T cells co-cultured with MenSC-derived secretomes. These findings demonstrate that the combination of IFNγ and TNFα represents an excellent priming strategy to enhance the immunomodulatory capacity of MenSCs. Moreover, the secretome derived from primed MenSCs may be postulated as a therapeutic option for the regulation of adverse inflammatory reactions.

## 1. Introduction

Human menstrual blood is an important source of stromal cells (MSCs). Menstrual blood-derived stromal cells (MenSCs) fulfill the minimum criteria established by the International Society for Cellular Therapy: they exhibit plastic adherence in tissue and can be passaged in standard culture media; they express a CD105+, CD73+, CD90+, CD45−, CD34−, CD14−, and HLA-DR− phenotype; and they possess an in vitro differentiation capacity toward osteoblasts, adipocytes, and chondroblasts [1]. MenSCs are also characterized by the well-known immunomodulatory and regenerative properties of MSCs, as well as immunosuppressive activity [2,3,4] and anti-apoptotic and pro-angiogenic capacities [5]. However, they stand out for their easy isolation, high proliferation rate, low immunogenicity, and lack of ethical issues compared with other sources of MSCs [6,7]. More importantly, MenSCs remain stable for at least 20 passages without mutations or visible abnormalities in vitro [1].

It is widely accepted that the activity of MSCs is mediated by paracrine factors contained in the released secretome [8]. MSC-derived secretomes are composed of a set of MSC-derived bioactive factors in soluble form, along with those encapsulated in extracellular vesicles (EVs). They comprise proteins, nucleic acids, and lipids that are crucial for intercellular communication and responsible for MSCs’ therapeutic potential [1]. However, defining if a single component exerts the therapeutic activity or if it is the result of a synergistic action is a controversial issue. Recent studies support the hypothesis that better effects are obtained with the use of the whole secretome instead of using only the exosome fraction [9,10].

These beneficial effects may be enhanced by appropriate preconditioning of the cells. Many strategies aiming to improve the regenerative effects and efficacy of released EVs have been recently proposed and reviewed [11,12]. Essentially, the main idea is to optimize the in vitro culture conditions and provide different stimuli (the so-called “priming” or “licensing”) to use MSCs as “cell factories”. Under these conditions, the “manufactured product” is different from the one produced under basal conditions. Nowadays, most of the priming strategies have been optimized, involving MSCs derived from bone marrow, umbilical cord, or adipose tissue and for disparate purposes: induction of angiogenesis, regenerative capacity, or enhancement of cell viability [12]. In this study, we have evaluated different priming strategies to enhance the immunomodulatory capacity of secretomes from MenSCs. The immunomodulation of CD4+ T cells by these secretomes was already demonstrated by our group [13], and transcriptomic and proteomic studies allowed us to understand the complexity of the molecular networks involved and their effects on the target cells [14].

Among the explored priming strategies, the effect of IFNγ and TNFα, alone or in combination, is noteworthy for EV-based therapies. EVs from IFNγ-primed MSCs have been found to enhance macrophage bacterial phagocytosis [15], and membrane particles obtained from IFNγ-primed MSCs can induce an increase in anti-inflammatory programmed death-ligand 1 (PD-L1) [16]. In the case of TNFα, EVs from TNFα-primed MSCs showed a therapeutic effect in urethral fibrosis [17], an improvement in the proliferation and differentiation of osteoblasts [18], as well as neuroprotective effects [19]. In the case of priming strategies using the combinations of IFNγ and TNFα, in vitro studies have revealed that IFNγ- and TNFα-primed MSCs produced EVs with a heightened immunomodulatory potential due to prostaglandin E2 and cyclooxygenase 2 pathway alteration [20]. Priming MSCs with IFNγ and TNFα also shifted macrophage polarization from M1 to M2 by the miRNAs [21] and resulted in immunomodulatory EVs, which induced M2 differentiation and the enhancement of Tregs [22]. Moreover, it has been demonstrated that IFNγ and TNFα priming upregulates indoleamine 2,3-dioxygenase (IDO1) [23], which is one of the major immunosuppression mechanisms of human MSCs [24]. IDO is involved in the L-tryptophan catabolism, leading to its depletion in the surrounding microenvironment and the accumulation of kynurenine, which then inhibits T cell activation and proliferation, among other effects [12]. All these studies suggest that IFNγ and TNFα have an enormous potentiality to act as priming agents for MSCs. However, it is important to note that these publications analyzed the effect of priming on cells but did not focus on the description of their secretomes or their secreted vesicles.

According to our previous studies [13,14], and considering all preceding findings, the first aim of this study was to evaluate different priming strategies in MenSCs to enhance their immunomodulatory capacity. Afterward, MenSC-derived secretomes were characterized, but the state-of-the-art techniques and their immunomodulatory properties were assessed. Our results first revealed a synergistic and additive effect of IFNγ and TNFα that significantly triggered the release of IDO1. Second, this priming strategy contributed to phenotypic changes in the molecules involved in migration, adhesion, and immunogenicity. Third, EV release was also boosted in IFNγ- and TNFα-primed MenSCs (MenSCs*). Four secretomes from the MenSCs* delivered miRNAs targeting inflammation-related genes. Finally, an improvement of the immunomodulatory capacity of the primed secretomes (S-MenSCs*) was demonstrated by in vitro functional assays.

To our knowledge, this is the first report focused on the optimization of priming strategies in MenSCs. Our results have demonstrated that IFNγ and TNFα priming has a profound impact on the release of EVs, miRNA profiles, and the immunomodulatory potential of these secretomes. Altogether, our results point out that these secretomes may have a future therapeutic use, with clinical implications in inflammatory-mediated diseases.

## 2. Results

### 2.1. Characterization of In Vitro Isolated and Expanded MenSCs

Our MenSCs were plastic-adherent in standard culture conditions, and their differentiation toward the adipogenic, chondrogenic, and osteogenic lineages was demonstrated by specific staining. A more detailed description of MenSC characterization can be found in our previous studies [13,14,25]. The phenotypic analysis of in vitro expanded MenSCs revealed a positive expression for the surface markers CD44, CD73, CD90, and CD105 and negative for CD14, CD20, CD34, CD45, CD80, and HLA-DR (data not shown). The cells used for this study complied with the minimal criteria for defining multipotent mesenchymal stromal cells [26].

### 2.2. Effect of MenSC Priming on IDO1 Release

To enhance the immunomodulatory capacity of secretomes from MenSCs, the effect of different pro-inflammatory cytokines (TNFα, IFNγ) and toll-like receptor (TLR) ligands (LPS, Poly (I:C)) was evaluated under in vitro conditions. The release of IDO1 was used as a primary biomarker for the immunomodulatory potential of MenSC secretomes. The ELISA quantifications demonstrated an increase in IDO1 release when the MenSCs were primed with 100 ng/mL of IFNγ. A significant increase was also observed when 100 ng/mL of IFNγ was combined with LPS or Poly (I:C) at 1 ng/mL. In the case of IFNγ at low concentrations (1 ng/mL), alone or in combination, the release of IDO1 was not increased when compared with the basal conditions. Interestingly, the combination of 100 ng/mL IFNγ and 100 ng/mL TNFα led to the highest level of IDO1 release (Figure 1). These results demonstrated a synergistic effect of IFNγ and TNFα on the immunomodulatory potential of MenSCs.

### 2.3. Phenotypic Profile of Surface Markers on MenSCs

The phenotype of the MenSCs (*n* = 3) and MenSCs* (*n* = 3) was analyzed by flow cytometry using a large panel of monoclonal antibodies toward various cell surface markers (Appendix A). Our phenotypic analysis revealed statistically significant differences in 9 out of 40 surface markers when the MenSCs and MenSCs* were compared (Figure 2a). All of them except CD49d were increased in the primed cells (Figure 2b). Moreover, these surface markers were classified in the following Gene Ontology (GO) categories: Immune System Process, Cell Adhesion, Cell Migration, and Response to Cytokine. On the other hand, the analysis by the Reactome Biological Pathways demonstrated a strong association with immune system-related categories (Figure 2c).

### 2.4. EV Secretion Profile of MenSCs and Phenotypic Analysis of Released Particles

To elucidate the EV contribution to the total secretome of MenSCs, we used nano-flow cytometry (nFC) to analyze the particle concentration and phenotype of the vesicles in the S-MenSC and S-MenSC* samples (Appendix A). To estimate the EV release rate, the particle concentration was directly determined by nFC in the secretome samples, and the data were normalized to the total viable cell number. Differences in cell viability were not observed between the MenSCs and MenSCs* (Appendix A). We noticed a significant increase in EV release up to threefold from the MenSCs* compared with the basal cells (*p* = 0.002) (Figure 3a). Notably, the median size of the secretome particles was similar between samples, consistent with the reported size of small EVs (40–200 nm) (Appendix A). To further characterize these particles, we performed size-exclusion chromatography (SEC) of the secretomes. This technique allowed us to separate the soluble fraction of the secretomes from the EVs (Appendix A). EV-enriched fractions from the SEC were pooled and subsequently analyzed by nFC. Size profiling indicated that both the basal and primed EVs were of a similar median size (63.67 ± 2.25 and 62.67 ± 1.61 nm, respectively) (Figure 3b) and of similar size distributions (Figure 3c). Interestingly, no differences in terms of size were observed between the analyses of the total secretome (Appendix A) and EV-enriched fractions (Figure 3b), supporting the high-resolution power of nFC and further confirming the increase in EV release by the MenSCs*. To characterize the phenotype of our EVs, three tetraspanin markers were evaluated by fluorescence in nFC. Both CD63 and CD81 were detected in all samples independent of the priming conditions (Figure 3d). In contrast, CD9 could be confirmed in all MenSC* samples but only in two out of three MenSC samples. A paired *t*-test showed no significant differences between the distinct cellular origins, although a tendency toward a higher frequency of CD9+ as well as CD81+ events was observed under IFNγ and TNFα priming compared with the basal conditions (Figure 3d).

### 2.5. Effects of IFNγ and TNFα Priming in the miRNome

Next-generation sequencing (NGS) was performed on the S-MenSCs (*n* = 3) and S-MenSCs* (*n* = 3). A total of 628 miRNAs were identified. The normalized and filtered expression values of counts per million (CPM) revealed that 40 of them (6.37%) were significantly different when the S-MenSCs and S-MenSCs* were compared (*p* value adjusted by Benjamini–Hochberg FDR correction ≤ 0.05) (Appendix A). The top 10 miRNAs with the highest fold change were hsa-miR-155-5p, hsa-miR-361-3p, hsa-miR-376a-3p, hsa-miR-424-3p, hsa-miR-27a-3p, hsa-miR-210-3p, hsa-miR-21-3p, hsa-miR-490-3p, hsa-miR-26a-2-3p, and hsa-miR-181a-5p.

Out of the 40 miRNAs significantly expressed, 25 were upregulated and 15 were downregulated in S-MenSCs*. Further analyses (unsupervised analysis including principal component analysis (PCA) and GO enrichment analysis) were performed in these miRNAs to address the differences in miRNome profiles between the secretomes. The results of the PCA indicated that each condition exhibited a unique miRNA profile, allowing separation into well-differentiated clusters (Figure 4a). The heat map-based unsupervised hierarchical clustering analysis showed the differences in the different miRNAs (Figure 4b) and corroborated the PCA analysis. Moreover, Gene Ontology enrichment analysis was performed to classify the miRNAs. For this analysis, miRNet and TAM 2.0 software were used. Several biological processes were found to be enriched, with the most relevant terms being Inflammatory Response (GO:0006954) (42.5%), apoptotic process (40%), and Immune Response (GO:0006955) (35%).

### 2.6. miRNA Target Prediction in Inflammatory Response

According to the miRNome results, the upregulated miRNAs were subsequently analyzed to determine the miRNA target network using miRTargetLink. Only interactions with strong experimental evidence were included. Our results demonstrated that 105 genes were identified as miRNA targets (a total of 6 miRNAs were excluded because of a lack of connections). An enrichment analysis, carried out with FunRich, revealed that 22 of the target genes were included in the GO category of Inflammation Response (GO:0006954). These genes were subclassified into four categories: Innate Immune Response, Adaptive Immune Response, Positive Regulation of Inflammatory Response, and Negative Regulation of Inflammatory Response (Figure 5).

### 2.7. Validation of miRNA Analyses by qPCR

Validation of the NGS results was performed by a quantitative polymerase chain reaction (qPCR) in both S-MenSCs (*n* = 3) and S-MenSCs* (*n* = 3) for a total of 14 miRNAs. Our analyses revealed that, despite the strong significances observed in the NGS data, none of the differences in miRNA expression between the S-MenSCs and S-MenSCs* were significant. Nonetheless, most of the trends were confirmed, except in hsa-miR-185-5p, hsa-miR-30b-5p, and hsa-miR-7-5p, whose expression in the S-MenSCs* was decreased. Based on the qPCR analyses, the highest differences were observed in hsa-miR-181a-5p, hsa-miR-34a-5p, and hsa-miR-210-3p. Moreover, hsa-miR-155-5p was only detected in the primed secretomes, and hsa-miR-146b-5p was undetectable in our qPCR assay (Appendix A).

### 2.8. Immunomodulatory Effect of S-MenSCs and S-MenSCs* on In Vitro Activated T Cells

To evaluate the immunomodulatory effect of secretomes against in vitro stimulated lymphocytes, peripheral blood lymphocytes (PBLs) from healthy human donors were stimulated with anti-CD2, anti-CD3, and anti-CD28 beads.

The immunomodulation assay demonstrated that the proliferation of CD4+ and CD8+ T cells was decreased when proliferating lymphocytes were co-cultured with secretomes. Notably, significant reductions in CD4+ T cell proliferation were observed with any concentration of S-MenSCs* (Figure 6a), while CD8+ T cell proliferation was significantly decreased when the lowest concentrations of S-MenSCs (20 and 40 µg/mL) or S-MenSCs* (20 µg/mL) were used (Figure 6b).

The immunomodulation assay was also focused on the regulation of immune checkpoints. The expression of these immune checkpoints was determined in CD4+ and CD8+ T cells. A significant increase in CTLA-4 was found in CD4+ T cells co-cultured with S-MenSCs (*p* = 0.0093) but not with S-MenSCs*(Figure 6c). Additionally, both the S-MenSCs and S-MenSCs* (80 µg/mL) significantly increased the expression of PD-1 on CD4+ T cells (*p* = 0.0015 and *p* = 0.0241, respectively) (Figure 6d). No significant differences were observed in CD8+ T cells (data not shown).

## 3. Discussion

The paracrine factors released by MSCs have become a promising tool for the regulation of adverse inflammatory events and to support regenerative processes. MSCs from menstrual blood (henceforth referred to as MenSCs) afford several advantages, as they can be obtained by non-invasive procedures from multiple donors and without ethical concerns. Isolation and expansion of these cells are easy and feasible, guaranteeing high growth rates in a relatively short time [1].

Recent theoretical developments have revealed that priming strategies can be used to enhance the immunomodulatory capacity of MSCs against multiple immune cells [11,12,27]. Even though some authors have suggested that primed cells may allow tumor development of pre-existing malignant cells [28], primed—or licensed—cells have mostly demonstrated efficacy [29] and a lack of toxicity [30].

Considering that secretomes from MenSCs have already exhibited an immunomodulatory behavior [13,14], for this study, it was of interest to investigate whether different priming strategies could further enhance their immunomodulatory potential.

Our first set of experiments was focused on the identification of the best priming strategy for MenSCs. The efficacy of the priming strategies of the study was evaluated in terms of IDO1 secretion by MenSCs. As a matter of fact, this enzyme is considered the “potency marker” for MSC-mediated immune suppression [31].

Our results have revealed that the concomitant use of IFNγ and TNFα, both at 100 ng/mL, led to a relevant increase of IDO1 secretion by MenSCs. This finding is in accordance with a previous study where the combination of IFNγ and TNFα as a priming strategy could trigger IDO upregulation, provoking the suppression of T cell proliferation and the differentiation of monocytes into M2 macrophages in vitro [23]. IDO upregulation may occur due to chromatin remodeling at the gene promoter [32] or via the IFNγ-JAK-STAT1 pathway [33], as already demonstrated for IFNγ-primed MSCs, but it is still unknown if TNFα can alter these mechanisms, inducing different activation pathways. Regardless, preconditioning MSCs with IFNγ and TNFα was demonstrated to trigger the release of EVs with an enhanced capacity for promoting M2 macrophage polarization [21,22], reducing the release of inflammatory cytokines in vitro [20] and enhancing regulatory T cells [22]. To the best of our knowledge, this is the first study where the synergistic effect of IFNγ and TNFα as a priming strategy was assessed in MSCs from menstrual blood.

It had already been evidenced that the combination of IFNγ and TNFα could change the expression of immunoregulatory molecules in MSCs [34]. Therefore, once the priming conditions were optimized, the next set of experiments was conducted to characterize the phenotype of IFNγ- and TNFα-primed MenSCs (MenSCs*). The comparative analysis between MenSC* and MenSC profiles revealed that 9 out of 40 surface markers were significantly modified in the MenSCs*. Among the upregulated adhesion molecules, there were the following: NCAM-1 (CD56), associated with MSC migration and homing [35]; ICAM-1 (CD54), involved in MSC homing [36] and dendritic cell immunomodulation [37]; LFA3 (CD58), which interacts with MSC ligands on T cells [38]; and integrin ITGA5 (CD49e), involved in cell-cell adhesion during active homing [39]. Considering that preclinical and clinical studies have demonstrated that the homing and migration of systemically administered MSCs is very low [40], the increased expression of adhesion molecules in IFNγ- and TNFα-primed cells may have a therapeutic implication by improving the migration capacity and engraftment of MSCs into inflammatory tissues. Notably, IL6R (CD126) was revealed to be induced by our IFNγ and TNFα priming, in contrast with previous studies where adipose-derived MSCs were preconditioned with IFNγ or TGFβ [41]. The expression of this receptor has been correlated with the osteogenic [42] and adipogenic [43] differentiation of bone marrow-derived MSCs. Here, we assumed that the increase of CD126 may have also reflected an increased susceptibility of MenSCs to IL6-mediated inflammation.

Our phenotypic analysis was also focused on the expression of the immune checkpoint receptors. Here, we demonstrated a significant increase of the molecules CTLA-4 (CD152) and PD-L1 (CD274) under IFNγ and TNFα priming. CTLA-4 expression was already reported to be released by MSCs under hypoxic conditions [44]. In turns, PD-L1 was already observed to be induced in MSCs under IFNγ and TNFα [45]. The inflammatory-mediated induction of these surface markers suggests that these molecules could be involved in the modulation of T cells and peripheral tolerance.

Regarding secretome characterization, we used nFC to demonstrate that both MenSCs and MenSCs* secrete EVs. This method provides an innovative and more precise approach to quantify EV preparations, showing higher sensitivity (>40 nm), among other advantages. In particular, the presence of EVs was further confirmed by the detection of tetraspanin positive events in the samples. No significant differences in the CD9+, CD63+, or CD81+ EV subpopulations between conditions were observed. Despite this, a tendency for higher-frequency CD9+ and CD81+ events was detected in EV preparations from the primed cells, which may provide mechanistic insights about the EV releasing process in MenSCs. In addition, EVs from both the MenSCs and MenSCs* had similar sizes, and when normalized to the total viable cell number, they showed a significant increase. This means that the secretory pathways could be altered by IFNγ and TNFα priming.

Consequently, being aware of the involvement of miRNAs in immunomodulatory pathways [14], we hypothesized that IFNγ and TNFα could trigger the release of miRNAs that target genes involved in inflammation and innate or adaptive immune responses. miRNAs have been widely described to be associated with EVs [46] but can also be found in a soluble form [47]. Moreover, the contribution of miRNA in the EV composition has been recently debated [48]. Therefore, we decided to study the miRNome of our secretomes.

The analysis by NGS allowed us to identify a set of significantly altered miRNAs following IFNγ and TNFα priming. This evidence indicates that alterations in the cell culture, such as the application of inflammatory priming conditions, have the capability of affecting miRNA loading and release, as already observed in secretomes derived from IFNγ-primed MenSCs [14]. Only the most relevant miRNAs identified by our NGS analysis will be described. First, hsa-miR-27a-3p appeared to be differentially expressed in IFNγ- and TNFα-primed MenSCs. This miRNA was already described to be carried by EVs from MSCs and to promote M2 macrophage polarization [49]. Furthermore, it targets *IFNG* and is involved in the regulation of *IRAK4*, a promoter of *NF-κB* [50], which regulates inflammatory and immune genes. Second, hsa-miR-185-5p was reported to alleviate the inflammatory response and reduce cell proliferation when EVs from MSCs were enriched with this miRNA [51]. Last but not least, we found a differential expression of hsa-miR-155-5p that targeted genes involved in Innate and Adaptive Immune Response, such as *CSF1R*, *ICAM1* or *BCL6*. This finding is directly in line with previous studies, where hsa-miR-155-5p expression resulted in enhancement in IFNγ-, as well as IFNγ- and TNFα-primed MSCs [52,53]. Additionally, the inhibition of miR-155-5p in rat bone marrow-derived MSCs was demonstrated to enhance the differentiation of T cells toward Th2 and Treg cells in vitro [54]. The main conclusion that can be drawn from our NGS analysis is that a variety of miRNAs in the secretome deriving from IFNγ and TNFα priming target the genes that are involved in the regulation of the inflammatory response. Thus, it may be speculated that the transfer of these miRNAs may have an impact on immune cell behavior and hence on the TH1 and TH2 balance.

Finally, our functional studies using in vitro stimulated lymphocytes demonstrated an inhibitory activity from these secretomes on lymphocyte proliferation. Although further studies need to be performed, it may be hypothesized that the differentially expressed miRNAs in S-MenSCs* could be involved in this inhibition. The immunomodulation assay also demonstrated a significant increase in immune checkpoints (PD-1 and CTLA-4) on CD4+T cells co-cultured with secretomes. Taking into account that MenSCs* exhibited an enhanced expression of CTLA-4 and PD-L1, we hypothesize that the soluble forms of these molecules could be involved in the immunomodulatory activity of secretomes. Of course, more experiments should be conducted to elucidate these hypotheses.

Taken together, our results demonstrated that IFNγ and TNFα had a synergistic effect on the secretion of IDO1 by MenSCs, being an efficient strategy for MSC licensing. Under this priming condition, the adhesion molecules, cytokine receptors, and immune checkpoint receptors showed a significant increase that may have altered their migration or adhesion to tissues, as well as their immunomodulatory activity. Finally, according to NGS analysis in the secretomes, EV characterization by nFC, and immune functional assays, we demonstrated that IFNγ and TNFα priming enhanced the release of miRNAs and vesicles with an increased immunomodulatory potential. In summary, secretomes derived from primed MenSCs may become a future therapeutic option in inflammatory diseases.

## 4. Materials and Methods

### 4.1. Isolation, Culture, and Characterization of Human MenSCs

A summary of all the experimental procedures is shown in Figure 7. Written informed consent was obtained from the human donors and approved by the ethics committee of the Jesús Usón Minimally Invasive Surgery Center (Cáceres, Spain). Human menstrual blood samples were collected from three healthy premenopausal women aged between 30 and 40 years without infections or immune disorders and not undergoing hormone therapy. MenSCs were isolated under sterile conditions according to a previously described protocol [13]. The isolated MenSCs were characterized by flow cytometry and differentiation assays, as in our previous studies [13,14,25].

### 4.2. Priming of MenSCs and IDO1 Quantification

Different lineages of in vitro cultured MenSCs (*n* = 3) were primed using the following recombinant pro-inflammatory cytokines and TLR ligands, either alone or in combination: IFNγ (Miltenyi Biotec Inc, Auburn, CA, USA) at 1 ng/mL and 100 ng/mL, TNFα (Miltenyi Biotec) at 1 ng/mL and 100 ng/mL, LPS (Sigma, Saint Louis, MO, USA) at 1 ng/mL, and Poly (I:C) (Miltenyi Biotec) at 1 ng/mL. On day 3, the supernatants from the in vitro primed MenSCs were collected to evaluate the release of IDO1. The levels of IDO1 were quantified by ELISA according to the manufacturer’s instructions (R&D SYSTEMS, Minneapolis, MN, USA).

### 4.3. Phenotypic Analysis and Comparison of MenSCs and MenSCs* Surface Markers

The phenotypic analysis of the cells was performed by flow cytometry using a panel of human monoclonal antibodies (Appendix A). The MenSCs at passages 5–7 and 80% confluence were cultured in vitro under basal conditions (MenSCs, *n* = 3) or with 100 ng/mL IFNγ and 100 ng/mL TNFα for 72 h (MenSCs*, *n* = 3). The MenSCs and MenSCs* were detached from the tissue culture flasks using PBS containing 0.25% trypsin and incubated with different antibodies for 30 min at 4 °C in PBS with 2% FBS. The cells were then washed, resuspended in PBS, and acquired in a FACSCalibur™ cytometer (BD Biosciences, Franklin Lakes, NJ, USA) equipped with CellQuest software (BD Biosciences). The detached cells were incubated with the Fluorescence Minus One Control (FMO control) to properly compensate for the flow cytometry data.

The phenotypic profiles of the MenSCs and MenSCs* were compared. Differentially expressed proteins underwent enrichment and biological pathway analyses with Cytoscape (version 3.7.2) [56], which includes the Gene Ontology Resource [57] and the Reactome Pathway Database [58].

### 4.4. Collection, Concentration, and Quantification of MenSC-Derived Secretomes

The MenSCs (*n* = 3) and MenSCs* (*n* = 3) (obtained as mentioned above) at passages 6–12 and 80% confluence were used for secretome collection. The cell culture medium (with or without IFNγ and TNFα) was replaced by DMEM with 1% penicillin/streptomycin, 1% glutamine, and 1% insulin-transferrin-selenium (Thermo Fisher Scientific, Waltham, MA, USA) after rinsing with PBS. After 96 h of cell culture at 37 °C and 5% CO_2_-conditioned media were collected, centrifuged first at 1000× *g* 10 min at 4 °C and again at 5000× *g* 20 min at 4 °C, and filtered through 0.45- and 0.22-µm mesh to eliminate dead cells and debris. The cells were detached with trypsin and counted with a Neubauer chamber. Viability was evaluated through Trypan blue staining (Thermo Scientific). Then, the condition media was concentrated using a 3-kDa MWCO Amicon^®^ Ultra device (Merck-Millipore, Burlington, MA, USA) by centrifugation at 4000× *g* for 1 h at 4 °C. The concentration of proteins from enriched secretomes was quantified by a Bradford assay (Bio-Rad Laboratories, Hercules, CA, USA). Finally, the S-MenSCs and S-MenSCs* were stored at −80 °C for further analyses.

### 4.5. Characterization of Extracellular Vesicles in MenSC-Derived Secretomes

EV release was determined in the secretome samples after 10,000× *g* centrifugation for 1 h to remove large particles. The viable cell number was used as a normalizer for the EV release rate. To enrich the EV fraction of the secretomes, SEC was performed (Appendix A). Briefly, 10 mL of Sepharose CL-2B (Thermo Scientific) was stacked into PS columns (Thermo Scientific), washed, and equilibrated with freshly filtered (0.22 µm) PBS. A volume of 0.5 mL of 10,000× *g* supernatants was loaded onto the column for fraction collection (0.5 mL per fraction and a total of 24 fractions were collected), which was started immediately using PBS as an elution buffer. EV-containing fractions 8–10 were pooled to homogenize the comparison between samples in further analyses. The protein concentration in the SEC fractions was estimated by the bicinchoninic acid assay (BCA) (Thermo Scientific). A representative SEC profile is shown in Appendix A. Samples were aliquoted and stored at −80 °C for subsequent analyses.

The particle concentration and size determination were analyzed by nFC. A NanoAnalyzer (NanoFCM Co. Ltd., Nottingham, UK) equipped with a 488-nm laser was calibrated using 200-nm polystyrene beads (NanoFCM Co. Ltd.) with a defined concentration of 2.08 × 10^8^ particles/mL. These beads were also used as a reference for particle concentration estimation. In addition, monodisperse silica beads (NanoFCM Co. Ltd.) of four different sizes were used as reference standards (68 nm, 91 nm, 113 nm and 155 nm) to calibrate the size of the EVs. Freshly filtered PBS was used for background subtraction. The total secretome samples (10k g supernatants) were analyzed using the appropriate negative control media. For data acquisition, the samples were diluted with filtered PBS, resulting in a particle count in the optimum range of 2000–12,000 events, and the data were collected for 1 min under a sample pressure of 1.0 KPa. The particle concentration and size distribution were calculated using NanoFCM software (NF Profession V1.08, NanoFCM Co. Ltd.).

For nFC fluorescence analyses, 2–6 × 10^8^ particles were resuspended in a final volume of 100 µL freshly filtered PBS. Particle immunostaining was performed overnight at 4 °C with 200 ng FITC-coupled antibodies against human CD9 (clone HI9a, Cat. 312104), CD63 (clone H5C6, Cat. 353006), and CD81 (clone 5A6, Cat. 349504; all obtained from BioLegend, Koblenz, Germany). Corresponding isotype antibodies were used as negative controls (200 ng, BioLegend). The unlabeled antibody was removed by 900 µL PBS washing and subsequent ultracentrifugation at 110,000× *g* for 100 min 4 °C in an Optima TM MAX-XP ultracentrifuge (TLA-45 fixed-angle rotor, Beckman Coulter, Inc, Brea, CA, USA). Stained and pelleted EVs were suspended in 50 µL of freshly filtered PBS. Data acquisition was performed as previously indicated for a fixed number of events (3000–5000). Immunostaining data from nFC (FCS 3.0 files) were exported and analyzed using FlowJoTM (v10) software. Staining was considered positive when the percentage of positive events showed a minimum fold change of two, relative to the matching isotype control.

### 4.6. miRNA Analysis by Next-Generation Sequencing (NGS)

miRNA sequencing experiments were performed at QIAGEN Genomic Services (QIAGEN, Hilde, Germany). The total RNA was isolated from 1 mL of each secretome using the exoRNeasy Serum/Plasma Kit (QIAGEN) according to the manufacturer’s instructions. During the sample preparation, QC Spike-ins were added as quality control. A total of 5 μL of purified RNA tagged with adapters containing a Unique Molecular Index (UMI) was converted into cDNA NGS libraries using the QIAseq miRNA Library Kit (QIAGEN). The cDNA was amplified in 22 cycles of PCR and purified. Library preparation QC was performed using a Bioanalyzer 2100 (Agilent Technologies, Santa Clara, CA, USA). Libraries were pooled in equimolar ratios, based on the quality of the inserts and the concentration measurements. They were then quantified by qPCR and sequenced on a NextSeq500/550 System (Illumina, Inc., San Diego, CA, USA) according to the manufacturer´s instructions. Raw data were converted to FASTQ files for each sample using bcl2fastq software version 2.20 (Illumina, Inc.). Quality control of the raw sequencing data was checked using the FastQCtool [59]. The data analyses, which included filtering, trimming, mapping, quantification, and normalization, were carried out using the CLC Genomics Workbench (version 20.0.2) and CLC Genomics Server (version 20.0.2). The human genome version GRCh38 was used as a reference database to annotate the miRNAs. The read sets were aligned to the reference sequences from miRbase (version 22). Due to the limited sample number, the power of the analysis was increased by strictly filtering out miRNAs with low counts. These provide little evidence of differential expression and interfere with some of the statistical approaches. In this data set, we chose to retain miRNAs if they were expressed at a CPM higher than 0.5 in at least two samples. The differential expression between secretomes was evaluated through Empirical Analysis of Differential Gene Expression (EDGE) analysis within the CLC bio.

Using Benjamini–Hochberg FDR-corrected *p* values, differentially expressed miRNAs at a significance level of 0.05 (FDR) were selected for unsupervised analysis (principal component analysis (PCA), clustering, and heat maps) using ClustVis (version 2.0) [60]. Enrichment analysis of these miRNAs was performed using miRNet (version 2.0) [61] and TAM (version 2.0) [62]. Moreover, significantly overexpressed miRNAs in S-MenSCs* were submitted to a miRTargetLink [63] analysis to determine the human target genes of these miRNAs. The results were filtered to form strong experimental evidence. Only genes included in the Gene Ontology category of Inflammation Response (GO:0006954) were taken into consideration. Enrichment analysis of these genes was performed using the FunRich: Functional Enrichment analysis tool (version 3.1.3) [64]. The target genes were subclassified into four categories: Innate Immune Response (GO:0045087), Adaptive Immune Response (GO:0002250), Positive Regulation of Inflammatory Response (GO:0050728), and Negative Regulation of Inflammatory Response (GO:0050729). Finally, a gene-miRNA interaction network with 35 nodes and 51 edges was built using Cytoscape. The datasets discussed in this publication have been deposited in Sequence Read Archive (SRA) data with the accession number PRJNA664968.

### 4.7. Validation of Transcriptomics Results by qPCR

To confirm the transcriptomic outcomes, 14 out of 25 upregulated miRNAs were selected for being directly involved in the Inflammatory Response category (GO:0006954) and were evaluated by qPCR. Briefly, the total RNA from the secretomes was isolated using a Total Exosome RNA and Protein Isolation Kit (Thermo Fisher), following the manufacturer’s protocol for total RNA extraction. An Implen NanoPhotometer™ spectrophotometer (Fisher Scientific) was used to quantify the purity and concentration of the total RNA. The cDNA was synthesized from 8 ng RNA in reverse transcription reactions performed with the TaqMan^®^ Advanced miRNA cDNA Synthesis kit (Thermo Fisher) according to the manufacturer’s instructions. For each sample, 6.5 µL of diluted cDNA (1:10) was used as a template in the qPCR amplification using a TaqMan ™ Fast Advanced Master Mix and TaqMan ™ Advanced miRNA Assay (Thermo Scientific) (Appendix A).

According to a previous study [65], two endogenous controls were selected for these experiments: has-miR-16-5p and has-miR-93-5p. Amplifications were run in triplicate in a QuantStudio 3 Real-Time PCR System (Applied Biosystems). Molecular biology-grade water replaced the cDNA in no-template control reactions. Amplifications later than the 37th cycle were considered to not be amplified. Following the normalization using the selected endogeny, the quantification of miRNAs was performed by 2^−ΔCrt^ calculation. As a final point, the 2^−ΔΔCrt^ value was calculated to analyze the difference between the control and primed conditions.

### 4.8. Immunomodulatory Assays on In Vitro Activated Peripheral Blood Lymphocytes

The immunomodulatory capacities of S-MenSCs (*n* = 3) and S-MenSCs* (*n* = 3) were determined on PBLs from healthy donors. Briefly, PBLs in RPMI-1640 medium and 10% FBS were stimulated with a T Cell Activation/Expansion Kit according to the manufacturer’s instructions (MACS, Miltenyi Biotec, Bergisch Gladbach, Germany) in U-bottom 96-well plates at 4 × 10^5^ cells per well. Secretomes were added to PBLs at different concentrations (20, 40, and 80 µg/mL). For the proliferation assays, PBLs (*n* = 4 donors) were previously stained with CellTrace™ CFSE (Thermo Scientific). On day 3, the in vitro activated PBLs were analyzed by flow cytometry. PBLs without stimulation were used as a negative control, and the in vitro activated PBLs without secretomes were used as a positive control. For phenotypic analyses of the immune checkpoints PD-1 and CTLA-4, in vitro activated PBLs (*n* = 3 donors) were collected at day 3 and incubated for 30 min at 4 °C with fluorescence-labeled human monoclonal antibodies against CD4, CD8, CTLA-4, and PD-1 (BD Biosciences) in PBS containing 2% FBS. The cells were then washed, resuspended in PBS, and acquired in a FACSCalibur™ cytometer (BD Biosciences). The flow cytometric analysis was performed on gated CD4+ T cells and CD8+ T cells. In the proliferation assays, the percentage of CFSE-low cells allowed us to quantify the proliferation of lymphocyte subsets.

### 4.9. Statistical Analysis

The data were statistically analyzed with GraphPad Prism (version 8.0) using a paired *t*-test for variables with parametric distribution and the Wilcoxon test for the non-parametric data. The data were considered statistically significant at *p* ≤ 0.05.

## Figures and Tables

**Figure 1 ijms-22-12177-f001:**
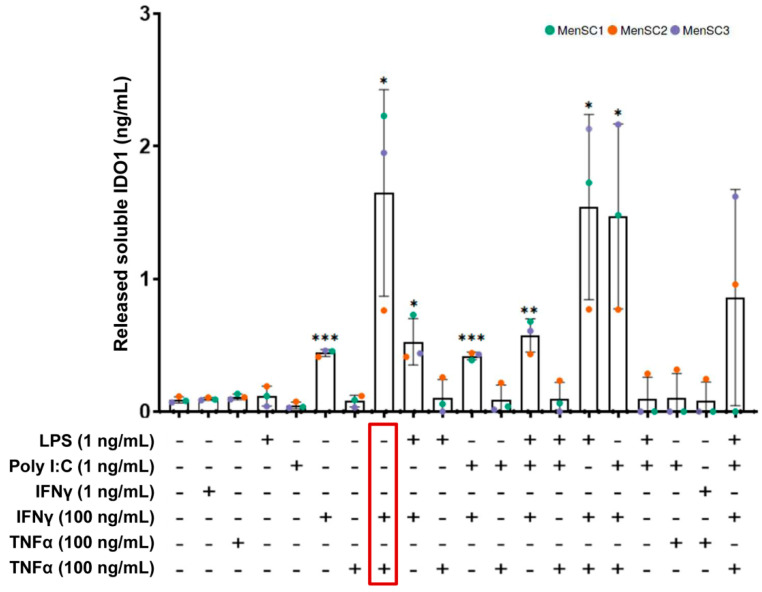
Release of soluble IDO1 by primed MenSCs. The MenSCs (*n* = 3) were primed with several combinations and concentrations of pro-inflammatory cytokines (IFNγ and TNFα) and the TLR ligands (LPS and Poly (I:C)). Levels of released IDO1 by the cells were tested by ELISA. A paired *t*-test was used to compare IDO1 levels in the basal MenSCs against different priming conditions. Colors refer to primary cultures from different donors. Error bars represent the standard deviations of data. Asterisks indicate statistically significant differences compared with the basal condition. * *p* < 0.05. ** *p* < 0.01. *** *p* < 0.001.

**Figure 2 ijms-22-12177-f002:**
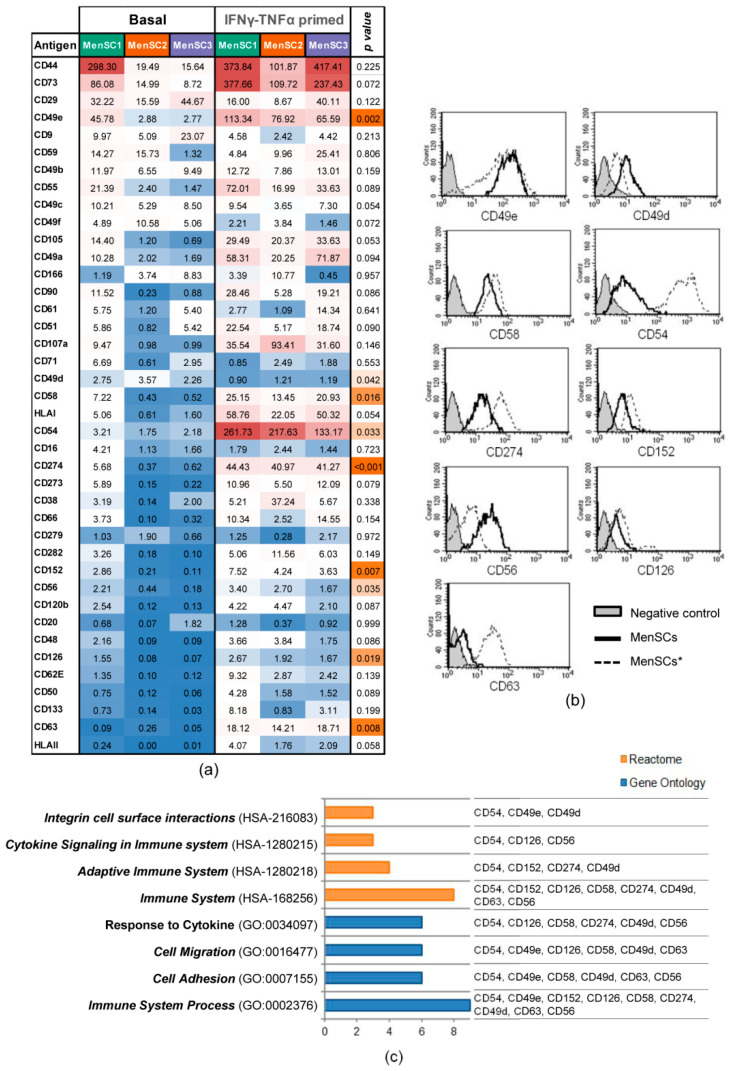
Phenotypic surface marker expression on MenSCs. Analysis of surface markers in basal MenSCs (*n* = 3) and MenSCs* (*n* = 3) over 3 days. (**a**) The expression levels of 40 cell surface markers (Stain Index (SI) value) are illustrated on a heat map. A paired *t*-test was carried out to compare MenSCs and MenSCs*, where *p* < 0.05 was considered statistically significant. The color scale for the SI values illustrates the highest (red) and the lowest (blue) expression values, and the orange color scale indicates the grade of signification, with the most significant ones in dark orange. (**b**) Representative histograms of statistically different markers. The expression in MenSCs is represented by a black-lined histogram, and the expression of MenSCs* is represented by a discontinuous line. The opaque gray histogram corresponds to the fluorescence negative control. (**c**) Reactome and Gene Ontology enrichment analyses of statistically significant different markers were performed by adjusting the *p* value with Benjamini–Hochberg FDR correction < 0.01. Graph bars represent the number of surface markers annotated into Reactome (orange) and Gene Ontology (blue) categories.

**Figure 3 ijms-22-12177-f003:**
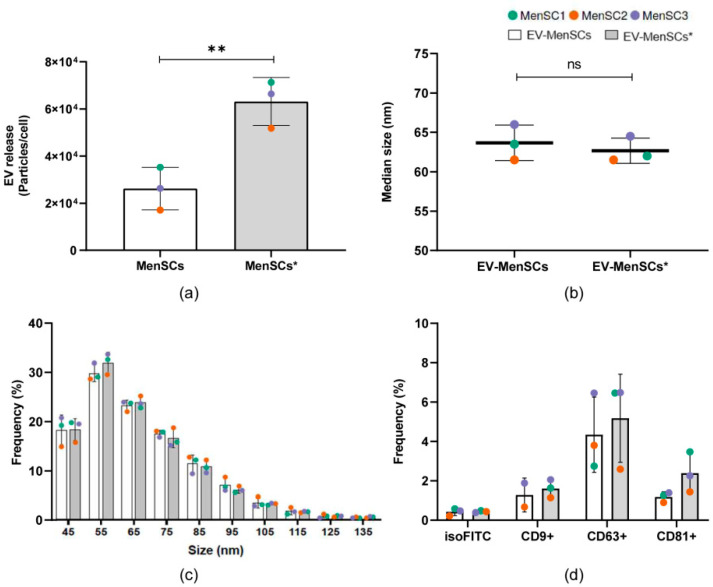
Characterization of the EV fraction in the secretome of MenSCs. (**a**) EV release rate of MenSC under basal (white bars) and priming (gray bars) conditions. The number of particles released by MenSCs after 96 h was determined for the secretome samples (10,000× *g* supernatants) by nFC. Values were normalized to the total number of viable cells. (**b**) Median size (nm) of particles detected in EV-enriched fractions (EV-MenSCs) obtained after size exclusion chromatography (SEC) of the secretomes. (**c**) Histogram of particle size (SEC samples) with a bin width of 10 nm from primed (gray bars) and basal (white bars) MenSCs. Mean ± SD of the relative frequency of total events detected by nFC is shown. (**d**) Single-particle phenotyping of EV samples obtained by SEC. Three tetraspanin markers (CD9, CD63, and CD81) were evaluated by single-vesicle fluorescence analyses. Only those samples which showed positive staining were plotted. Different colored dots represent the values obtained with independent donors (*n* = 3). Significance was tested by a paired *t*-test. ** *p* < 0.01 in primed vs. basal conditions.

**Figure 4 ijms-22-12177-f004:**
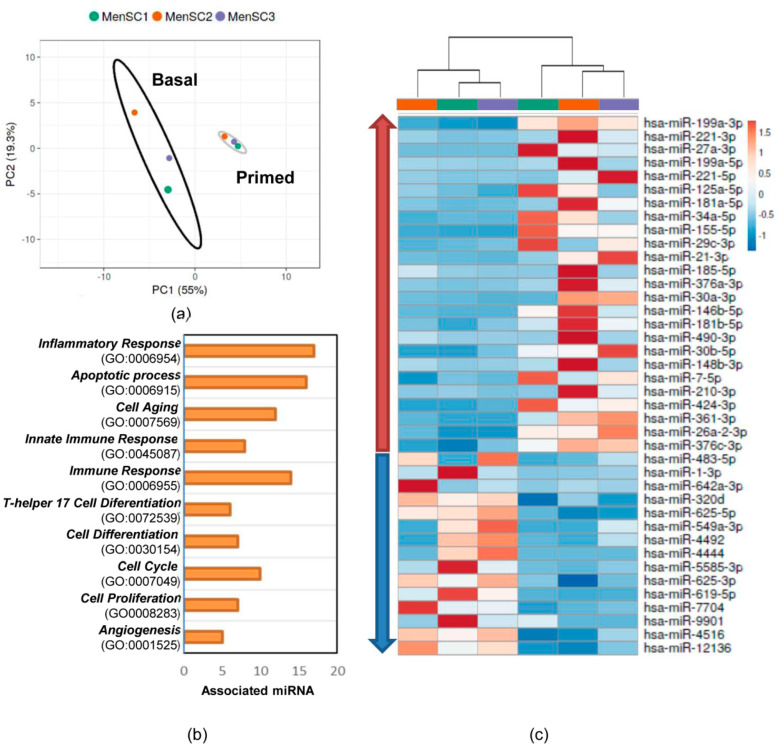
Effects of IFNγ and TNFα priming on the miRNome of S-MenSCs. The 40 significantly regulated miRNAs in the S-MenSCs* were evaluated. (**a**) Principal component analysis (PCA) plots, with the score plot for PC1 (55% variance explained) vs. PC2 (19.3% variance explained). (**b**) Hierarchical clustering of the primary cells (MenSC1, MenSC2, and MenSC3) under different conditions (basal and IFNγ and TNFα priming) together with the heat maps of miRNAs. (**c**) Enrichment analysis revealed a significant implication of the selected miRNAs in the Gene Ontology categories (*p* value adjusted by Benjamini–Hochberg FDR correction < 0.01). Graph bars represent the number of associated miRNAs in each category.

**Figure 5 ijms-22-12177-f005:**
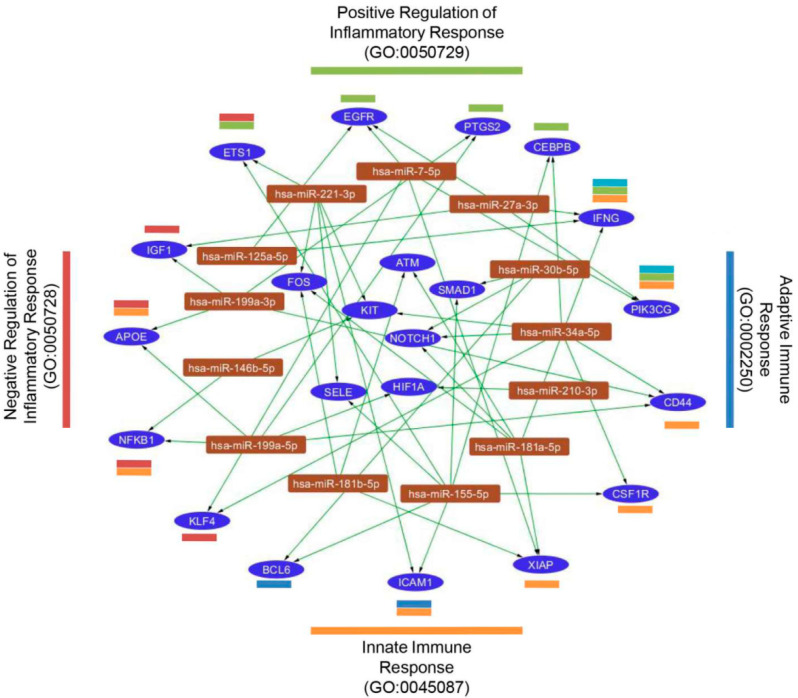
Predicted target genes for miRNAs. Target genes for significantly increased miRNAs on primed MenSC secretomes were analyzed by miRTargetLink. Target genes were first classified into the category Inflammatory Response (GO:0006954) and then subclassified into four subcategories (Innate Immune Response (orange), Adaptive Immune Response (blue), Positive Regulation of Inflammatory Response (green), and Negative Regulation of Inflammatory Response (red)). The interaction network of gene targets (blue ellipses) and miRNAs (brown rectangles) was illustrated using Cytoscape.

**Figure 6 ijms-22-12177-f006:**
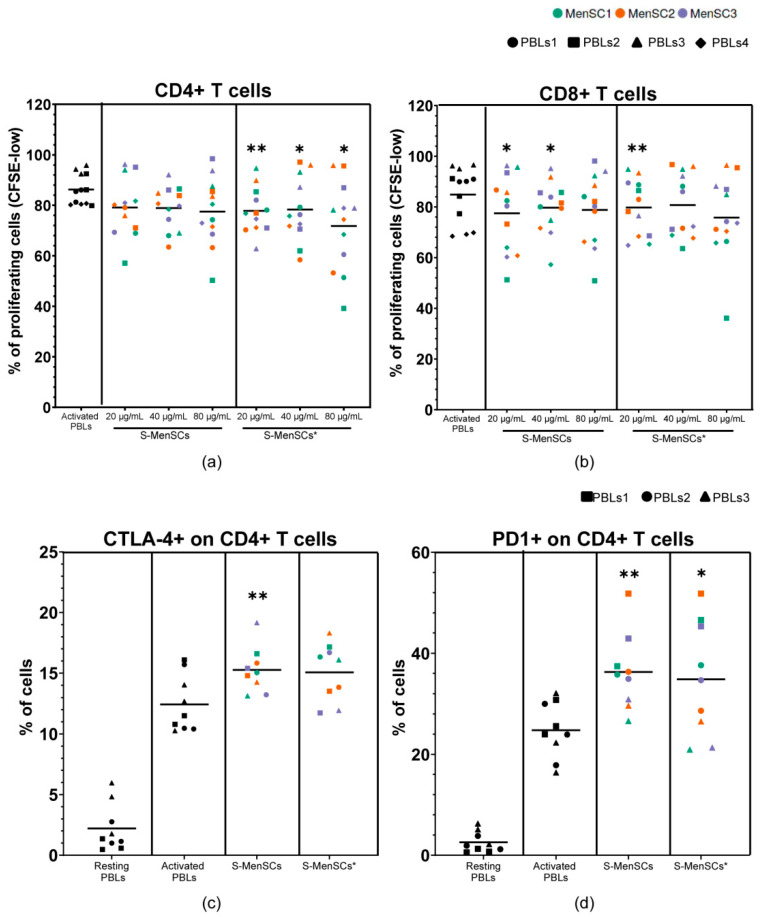
Lymphocyte activation assays for the evaluation of the immunomodulatory effect of S-MenSCs and S-MenSCs*. PBLs from healthy donors (*n* = 3–4, different symbols) were activated and co-cultured with S-MenSCs and S-MenSCs* for 72 h (*n* = 3, different colors). (**a**) Percentage of CD4+ cells and (**b**) percentage of CD8+ T cells with CFSE-low. (**c**) Percentage of CTLA-4 expression on CD4+ T cells. (**d**) Percentage of PD-1 expression on CD4+ T cells. Individual values are represented together with the mean. Data were analyzed by paired *t*-test against activated PBLs as matched controls. A *p* value was considered significant at * *p* < 0.05, ** *p* < 0.05.

**Figure 7 ijms-22-12177-f007:**
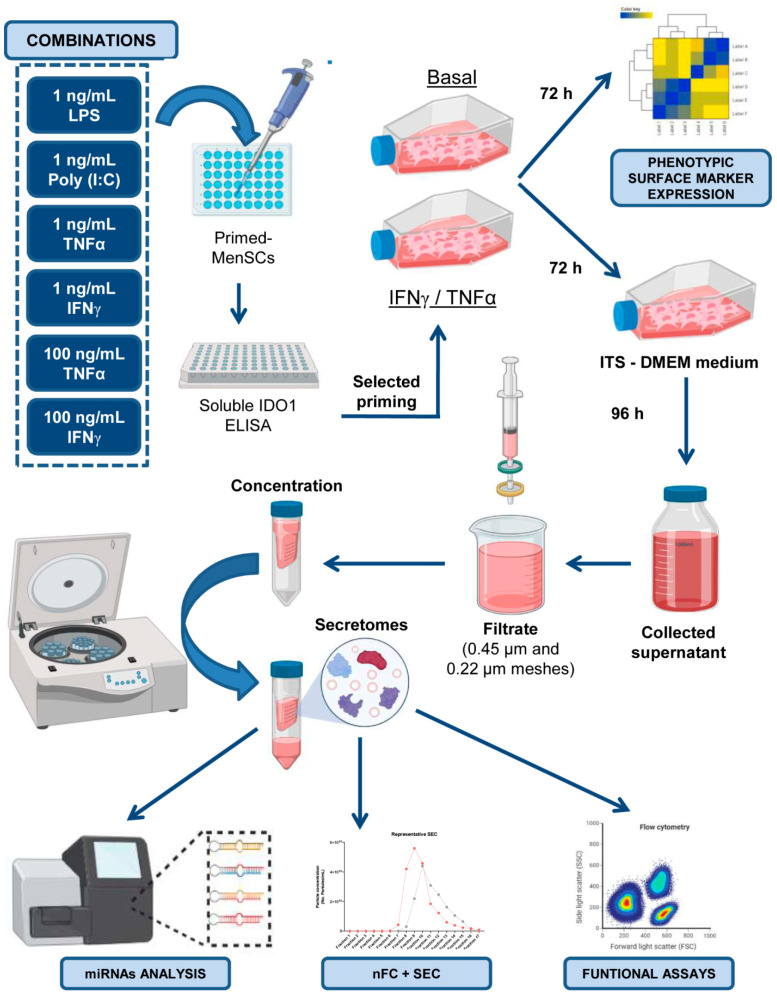
Graphic overview of key experimental procedures. LPS, Poly (I:C), TNFα, and IFNγ were used at different concentrations, individually or in combination, to prime in vitro cultured MenSCs. The optimal priming strategy was identified by ELISA in terms of IDO1 release. For the ELISA results, the MenSCs were primed with 100 ng/mL IFNγ and TNFα, and the phenotypic profiles of MenSCs* and MenSCs were determined by flow cytometry. Moreover, the MenSCs and MenSCs* were cultured with ITS (insulin-transferrin-selenium)-DMEM medium serum-free. After 96 h, conditioned media were concentrated to obtain the total secretome fraction, which was also enriched in extracellular vesicles. Finally, these secretomes were characterized by next-generation sequencing (NGS), nano-flow cytometry in combination with size exclusion chromatography (nFC+ SEC), as well as with lymphocyte activation assays. Images have been created with BioRender [55].

## Data Availability

The datasets generated for this study can be found in the Sequence Read Archive (SRA) data with the accession number PRJNA664968 (http://www.ncbi.nlm.nih.gov/bioproject/664968).

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
