# Peer review of "IFN-Gamma and TNF-Alpha as a Priming Strategy to Enhance the Immunomodulatory Capacity of Secretomes from Menstrual Blood-Derived Stromal Cells"

_ijms, 2021, doi:10.3390/ijms222212177_

Round 1
Reviewer 1 Report
In this study the authors aimed to investigate different priming strategies to increase the immunomodulatory capacity of the MenSCs. The study was well conducted, the results are clearly described, and the data obtained are interesting.
The work can be accepted
Author Response
Thank you for your kind comments.
Reviewer 2 Report
This manuscript by María Ángeles de Pedro et al showed that IFN-r and TNFα primed MenSC exhibited enhanced IDO1 expression, increased EV secretion and identify some miRNA that showed regulatory effects on immune system. They also found IFNr/TNFα primed MenSCs showed decrease CD4+T cells and CD8+T cells and increase CTLA4 and PD1 in CT4+T cells and conclude that these IFNr and TNFα primed MenSC showed immunoregulatory effect. A previous study showed MenSCs’ immunosuppressive effects are lower than BM-MSCs. When primed with IFN-r and Il-1β, but showed superior protective effect on in vivo GVHD model independent of immune regulation (26528946). Therefore, an in vivo study will be more desired for the future direction.
Overall, this in vitro study manuscript is well written and fit IJMS.
Few minor comments.
- Figure 1 legend: Statistic stars should point out compared to which group.
- Figure 3, Histogram legend showed both bar chart color and cell populations. It is confusing.
- There is period before and after parenthesis of figure, Line 243.
- Some language tended like oral English such as “All things considered in the conclusion section”. People usually write: Take together, or in summary”.
Author Response
Response to Reviewer 2 Comments:
Figure 1 legend: Statistic stars should point out compared to which group.
In order clarify this point, we have completed the sentence as follows:
“Asterisks indicate statistically significant differences with the basal condition: *p < .05, **p < .01, *** p < .001.”
Figure 3, Histogram legend showed both bar chart color and cell populations. It is confusing.
According to the reviewer's comment, we added colored dots to the panels and a new Figure 3 (c) has been included in the manuscript.
There is period before and after parenthesis of figure, Line 243.
Thanks for your comment. The extra period has been deleted.
Some language tended like oral English such as “All things considered in the conclusion section”. People usually write: Take together, or in summary”.
Thanks to the reviewer's suggestion, we have replaced our sentence with “Taken together” at the beginning of the conclusions section and by “In summary” in the last sentence of the manuscript.
Further minor changes:
- New affiliation has been included.
- English grammar revisions.
